# Non-unitary dynamics of Sachdev-Ye-Kitaev chains

Chunxiao Liu,[1] Pengfei Zhang,[2, *] and Xiao Chen[3, †]

[1]*Department of Physics, University of California Santa Barbara, Santa Barbara, California, 93106, USA*
[2]*Institute for Quantum Information and Matter and Walter Burke Institute for Theoretical Physics,*
*California Institute of Technology, Pasadena, California 91125, USA*
[3]*Department of Physics, Boston College, Chestnut Hill, MA 02467, USA*

(Dated: September 7, 2020)

We construct a series of one-dimensional non-unitary dynamics consisting of both unitary and imaginary evolutions based on the Sachdev-Ye-Kitaev model. Starting from a short-range entangled state, we analyze the entanglement dynamics using the path integral formalism in the large $N$ limit. Among all the results that we obtain, two of them are particularly interesting: (1) By varying the strength of the imaginary evolution, the interacting model exhibits a first order phase transition from the highly entangled volume law phase to an area law phase; (2) The one-dimensional free fermion model displays an extensive critical regime with emergent two-dimensional conformal symmetry.

## CONTENTS

## 1. INTRODUCTION

Recent years have witnessed tremendous breakthrough in many-body quantum dynamics. For a closed many-body quantum system decoupled from the environment, under the unitary dynamics, the interaction in the system can lead to chaos and thermalize all the small subsystems. The total wave function acts as its own heat bath and this phenomenon is referred as quantum thermalization [1, 2].

The irreversible thermalization process can be avoided if we allow non-unitary evolution, which naturally arises in open quantum systems. Recently it is observed that a unitary dynamics subjected to repeated measurement can

_____________

\* PengfeiZhang.physics@gmail.com
† chenaad@bc.edu

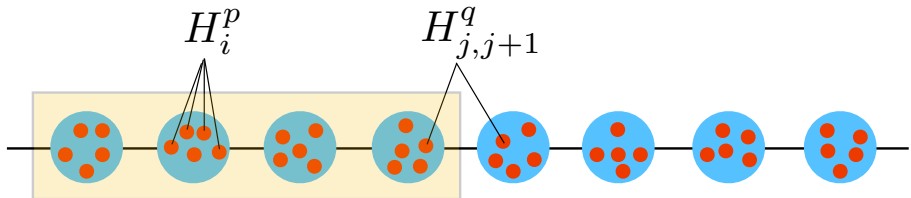

Figure 1. The cartoon for the non-Hermitian SYK model. The $H_i^p$ represents the intra-cluster SYK model and the $H_{j,j+1}^q$ represents the inter-cluster SYK coupling term. We compute Rényi entropy of a single interval in the box.

exhibit non-thermal phases if we follow the *quantum trajectory* of the many-body wave function. More strikingly, by varying the measurement rate, there is a continuous entanglement phase transition [3–17]. In the phase with slow measurement rate, the state remains highly entangled and the entanglement entropy obeys volume law scaling, while in the phase with fast measurement rate, the entanglement entropy obeys area law scaling. Notice that the explicit form of the measurement is not important and can be either a projective measurement or a more generalized weak measurement [5, 15].

Motivated by these findings, we consider the following question: For a non-unitary dynamics $U \sim \exp(-iHt)$ governed by a non-Hermitian Hamiltonian

$$H = H_1 - igH_2, \tag{1}$$

can we realize an entanglement phase transition by varying $g$? In the above equation, both $H_1$ and $H_2$ are Hermitian Hamiltonians. More specifically, in this paper, we consider the following interaction: $H_1$ is a Hamiltonian describing interaction between different sites and $H_2$ is a Hamiltonian defined at each site. $H_2$ for example can describe the coupling of the onsite degrees of freedom to an external field. For such non-unitary dynamics, in the limit $g = 0$, we expect that the steady state will typically saturate to a highly entangled state with a volume law scaling, while in the limit $g \to \infty$, this becomes a purely imaginary evolution and the steady state is a trivial product state with zero entanglement entropy. In a strongly interacting system, it is not obvious if there is a phase transition occurring at finite $g$. To address the above questions, we consider a one-dimensional (1D) non-unitary dynamics constructed from Sachdev-Ye-Kitaev (SYK) model [18, 19] and explore the possible phase transition in it.

The SYK model is a fermionic system with random all-to-all interaction [18–20]. This model can be analytically solved in the large $N$ limit. Due to this special property, this model has vast applications in different fields including high energy, condensed matter physics and quantum information theory. Many variants of the SYK model have been constructed to study quantum chaos, quantum gravity and non-Fermi liquid analytically [21–30]. In particular, there are studies on the entanglement entropy of the SYK model [31–37], where transitions to the replica wormhole [38, 39] solution are found in the long time limit.

In this paper, we will use the SYK model to construct a set of 1D chain models and explore the non-unitary dynamics harbored in them. We study the entanglement dynamics by using the path integral formalism, which can be obtained self-consistently by virtue of the large $N$ nature of the SYK model. By varying $g$, we observe different entanglement scaling behaviors which correspond to different saddle point solutions. We hope these results in the large $N$ limit could shed light on the more generalized phase transition in interacting systems at finite $N$ where analytical tools are lacking.

The rest of the paper is organized as follows. In Sec. 2 we define the SYK chain models, derive the path integral formalism for the Rényi entropy of these models, and write down the corresponding saddle point equations which are amenable to numerical study. Then in Sec. 3 we apply the formalism to two models: the interacting model with inter-cluster SYK$_4$ coupling and intra-cluster SYK$_2$ coupling, and the non-interacting model with inter-cluster SYK$_2$ coupling and intra-cluster SYK$_2$ coupling. We find that the steady state of the interacting model exhibits either a volume-law or an area-law phase as the coupling $\lambda$ is varied, and the two phases are separated by a first order transition, while that of the non-interacting model exhibits critical behavior for any finite coupling $\lambda$. We finally summarize and discuss these results in Sec. 4. Some derivation details, as well as results for another model with both inter- and intra-cluster SYK$_4$ interactions, are given in the Appendices.

## 2. MODEL AND METHOD

We consider non-unitary time evolution of SYK chain generated by the following non-Hermitian Hamiltonian:

$$H \equiv J H^p_{\text{inter-cluster}} - iV H^q_{\text{intra-cluster}} \equiv J \sum_x H^p_{x,x+1} - iV \sum_x H^q_x, \tag{2}$$

where $H^p_{\text{inter-cluster}}$ is the inter-site interaction with coupling strength $J$ and $H^q_{\text{intra-cluster}}$ is the onsite interaction with coupling strength $V$. From now on, we drop their subscript for conciseness. At each site, there are $N$ Majorana fermions. $H^p_{x,x+1}$ denotes $p$-body random interaction between neighboring sites $x$ and $x+1$ and $H^q_x$ is $q$-body random interaction at site $x$ (See the cartoon in Fig. 1). Their explicit forms are

$$H^p_{x,x+1} = i^{\frac{p}{2}} \sum_{i_1 < i_2 < \cdots < i_{p/2}, j_1 < j_2 < \cdots < j_{p/2}} J^{x,x+1}_{i_1 i_2 \cdots i_{p/2} j_1 j_2 \cdots j_{p/2}} \chi^x_{i_1} \chi^x_{i_2} \cdots \chi^x_{i_{p/2}} \chi^{x+1}_{j_1} \chi^{x+1}_{j_2} \cdots \chi^{x+1}_{j_{p/2}}, \tag{3}$$

$$H^q_x = i^{\frac{q}{2}} \sum_{i_1 < i_2 < \cdots < i_q} V^x_{i_1 i_2 \cdots i_q} \chi^x_{i_1} \chi^x_{i_2} \cdots \chi^x_{i_q.} \tag{4}$$

where the subscript $i_m, j_n = 1, 2, \ldots, N$ is the Majorana flavor index at each site and the superscript $x = 1, 2, ..., L$ is the site of the SYK chain; we assume periodic boundary condition for the chain so that $L + 1 \equiv 1$. $J^{x,x+1}_{i_1 \cdots i_{p/2} j_1 \cdots j_{p/2}}$ and $V^x_{i_1 i_2 \cdots i_q}$ are time-independent Gaussian random variables with vanishing mean and the following variance

$$\overline{|J^{x,x+1}_{i_1 \cdots i_{p/2} j_1 \cdots j_{p/2}}|^2} = \frac{(p/2)!(p/2-1)!}{2N^{p-1}}, \qquad \overline{|V^x_{i_1 i_2 \cdots i_q}|^2} = \frac{(q-1)!}{N^{q-1}}. \tag{5}$$

Note that both $H^p_{x,x+1}$ and $H^q_x$ are Hermitian Hamiltonian. Therefore in the time evolution governed by $\exp(-iHt)$, $H^p_{x,x+1}$ provides the unitary evolution while $H^q_x$ is responsible for the imaginary evolution. We describe the path integral formalism for this non-unitary evolution in the next subsection.

### 2.1. Rényi entropy under non-unitary evolution

We are interested in the entanglement dynamics of certain pure state. We prepare an initial state $|\psi(t = 0)\rangle = \otimes^L_{x=1} |\{0\}\rangle_x$ which is the state annihilated by all the complex fermions defined by $c_{x,j} = \chi_{x,2j-1} + i\chi_{x,2j}$ as $c_{x,j}|\{0\}\rangle_x = 0$, for all $x \in 1, 2, ..., L$ and $j = 1, 2, ..., N$. Since our initial state is a product state between different sites, there is no spatial entanglement.

We then evolve $|\psi(t = 0)\rangle$ under the Hamiltonian (2) to entangle different sites. At time $t$, the state reads

$$|\psi(t)\rangle = \frac{1}{\sqrt{Z[\{J^{x,x+1}_{i_1 \cdots i_{p/2} j_1 \cdots j_{p/2}}, V^x_{i_1 i_2 \cdots i_q}\}](t)}} e^{-itH}|\{0\}\rangle = \frac{1}{\sqrt{Z[\{J^{x,x+1}_{i_1 \cdots i_{p/2} j_1 \cdots j_{p/2}}, V^x_{i_1 i_2 \cdots i_q}\}](t)}} e^{-itJH^p - tVH^q}|\{0\}\rangle, \tag{6}$$

where

$$Z[\{J^{x,x+1}_{i_1 \cdots i_{p/2} j_1 \cdots j_{p/2}}, V^x_{i_1 i_2 \cdots i_q}\}](t) = \langle\{0\}|e^{itH^\dagger} e^{-itH}|\{0\}\rangle \tag{7}$$

is introduced to ensure normalization of the state $|\psi(t)\rangle$. From now on, we keep the dependence of random parameters implicit. It is useful to write $\sqrt{Z(t)}|\psi(t)\rangle$ and $Z(t)$ as path-integrals:

$$\int_{\text{b.c.}} D\chi^x_i(\tau) \exp\left(-\int d\tau \left\{\frac{1}{2} \sum_{x,i} \chi^x_i \partial_\tau \chi^x_i + \sum_x V H^q_x[\chi^x_i] + f(\tau) \sum_x J H^p_{x,x+1}[\chi^x_i]\right\}\right) \tag{8}$$

Here we introduce $f(\tau)$ to specify whether the real-time evolution is forward or backward: $f(\tau) = i$ for forward evolutions $e^{-iH^p t}$ and $f(\tau) = -i$ for backward evolutions $e^{iH^p t}$ [1]. We use b.c. to denote proper boundary conditions

---

[1] Note that the direction of real-time evolution is contour-dependent, i.e $f(\tau)$ is a contour-dependent quantity.

specified by the pictorial representations:

$$\sqrt{Z(t)}|\psi(t)\rangle = \left\{ \begin{array}{c} \text{[diagram]} \end{array} \right\}, \qquad Z(t) = \left\{ \begin{array}{c} \text{[diagram]} \end{array} \right\} \tag{9}$$

where we have separated out the Majorana modes with even/odd indices as $\chi_{e/o}^x$ and $x \in [1, L]$. The solid lines represent the evolution and the black dots denote the initial state. The dotted lines represent interaction between fermions, which contains coupling between different sites. For $\sqrt{Z(t)}|\psi(t)\rangle$, additional quantum state is attached to the free ends.

We are interested in the Rényi entropy of $|\psi(t)\rangle$. We divide the SYK chain into subsystems $A$ and $B$, with $A$ containing sites $x = 1, 2, ..., L_A$. The reduced density matrix $\rho_A = \text{Tr}_B \rho(t)$ is obtained by tracing over the degrees of freedom in $B$. A pictorial representation of $\rho_A$ reads:

$$Z(t)\rho_A(t) = \left\{ \begin{array}{c} \text{[diagram]} \end{array} \right\} \tag{10}$$

Here the red line represents the $A$ subsystem and the blue line represents the $B$ subsystem. The $n$th Rényi entropy is then defined as

$$S^{(n)} = -\frac{\log \text{Tr}_A \rho_A^n}{n-1} = -\frac{\log \left( Z_n(t, L_A)/Z^n(t) \right)}{n-1}, \tag{11}$$

where $Z_n(t, L_A)$ is given by sewing $n$ copies of reduced density matrix $\rho_A$, as shown in Fig. 2, where we have labeled contour $\mathcal{C}$ anti-clockwise by a single time variable $s \in [0, 4nt]$ (See Appendix A for details). The contour itself again defines the path-integral for computing the Rényi entropy, with boundary conditions indicated by dashed lines and black dots. From the contour, it is obvious that there is a symmetry by interchanging $A$ and $B$: $S^{(n)}(t, L_A) = S^{(n)}(t, L - L_A)$. In addition, we have $Z_n(t, 0) = Z_n(t, L) = Z^n(t)$, which is consistent with the fact that the full system is in a pure state.

To proceed, we need to average over random variables $J_{i_1...i_{p/2}j_1...j_{p/2}}^{x,x+1}$ and $V_{i_1...i_q}^x$ after computing $\log \text{Tr}_A \rho_A^n$ for each realization of $J_{i_1...i_{p/2}j_1...j_{p/2}}^{x,x+1}$ and $V_{i_1...i_q}^x$. Additional disorder replicas are required to accomplish this. In the SYK models that we consider, however, the computation can be simplified by working within a disorder replica diagonal ansatz: both numerical analysis and analytical arguments [40–43] suggest that at leading order of $1/N$, the Rényi entropy can be approximated by

$$S^{(n)} = -\frac{\overline{\log \text{Tr}_A \rho_A^n}}{n-1} \simeq -\frac{\log \overline{\text{Tr}_A \rho_A^n}}{n-1} = -\frac{1}{n-1} \left( \log \overline{Z_n(t, L_A)} - \log \overline{Z_n(t, 0)} \right), \tag{12}$$

and the result becomes exact in the large $N$ limit.

After disorder average, to compute $\overline{Z_n(t)}$ one could further introduce the bilocal fields $G_x$ and $\Sigma_x$ [20, 36]. Leaving details for Appendix A, we find

$$\overline{Z_n(t)} = \int_{\mathcal{C}} DG_x D\Sigma_x e^{-NS_n[G_x, \Sigma_x]}, \tag{13}$$

with

$$S_n[G_x, \Sigma_x] = -\frac{N}{4} \sum_x \left\{ \log \det \left[ \partial_{s,x} - \Sigma_x \right] - \int_0^{4nt} ds \int_0^{4nt} ds' \left[ \Sigma_x G_x - f(s)f(s') \frac{J^2}{p} \left( G_x G_{x+1} \right)^{\frac{p}{2}} P - \frac{V^2}{q} G_x^q P \right] \right\}. \tag{14}$$

As aforementioned, the time parameters $s, s' \in [0, 4nt]$ are parameterized on the contour $\mathcal{C}$. $P(s, s')$ is defined as

$$P(s, s') = \begin{cases} 1, & s, s' \text{ both parameterizing even or odd fields,} \\ 0, & s \text{ parameterizing odd (even) fields and } s' \text{ parameterizing even (odd) fields.} \end{cases} \tag{15}$$

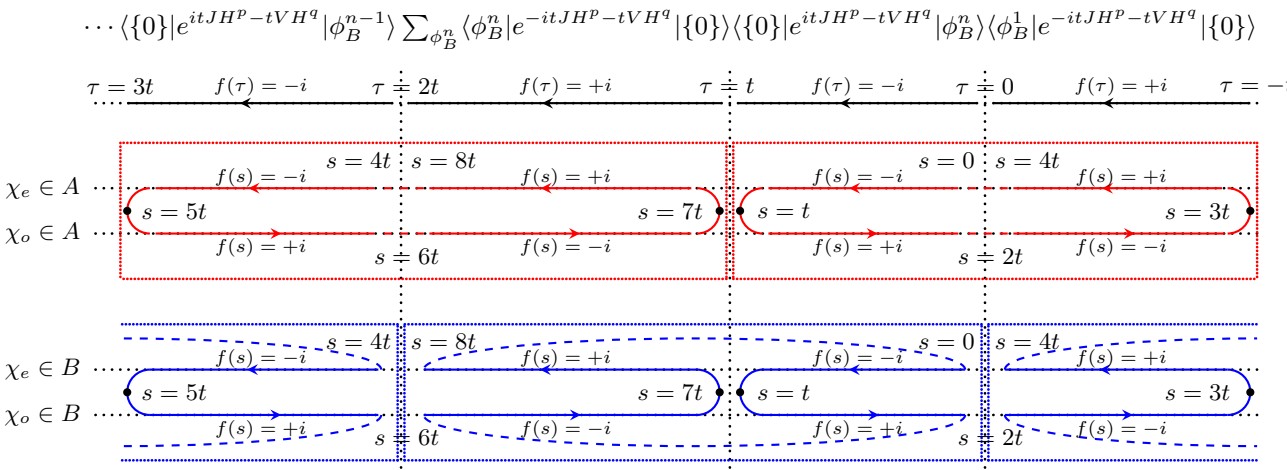

Figure 2. Time contour $\mathcal{C}$ (solid red and blue lines) parameterized by $s$ on the domain $s \in [0, 4nt]$ used in Eq. (14), which combines the $n$ replicas. The boundary conditions are shown by the dashed lines. The contour inside a blue box defines the Majorana fields of subsystem $B$ in one replica, while the contour inside a red box defines the Majorana fields shifted by the twist field (See Appendix B). $f(s) = \pm i$ indicates the direction of the real-time evolution (forward/backward) on the contour for $\chi(s)$. The original time parameterization by $\tau \in [0, 2nt]$ is also shown (solid black line) for comparison (See Appendix A).

In the large $N$ limit, since the action (14) is proportional to $N$, we obtain the saddle point solution

$$G_x(s, s') = \left(\partial_{s,x} - \Sigma_x^J - \Sigma_x^V\right)^{-1}(s, s'), \tag{16a}$$

$$\Sigma_x^J(s, s') = \frac{J^2}{2} f(s) f(s') G_x^{\frac{p}{2}-1}(s, s') \left(G_{x+1}^{\frac{p}{2}}(s, s') + G_{x-1}^{\frac{p}{2}}(s, s')\right) P(s, s'), \tag{16b}$$

$$\Sigma_x^V(s, s') = V^2 G_x^{q-1}(s, s') P(s, s'). \tag{16c}$$

Therefore the Rényi entropy is

$$S^{(n)} = \frac{1}{n-1} \left(S_{n,\text{saddle}}(L_A) - S_{n,\text{saddle}}(0)\right), \tag{17}$$

where $S_{n,\text{saddle}}$ is the on-shell action (18) obtained from the solution of the Eqs. (16). Explicitly, we have

$$S_{n,\text{saddle}}[G_x, \Sigma_x] = -\frac{N}{4} \sum_x \left\{\log \det\left[\partial_{s,x} - \Sigma_x\right] - \int_0^{4nt} ds \int_0^{4nt} ds' \left[\frac{p-1}{p} \Sigma_x^J G_x + \frac{q-1}{q} \Sigma_x^V G_x\right]\right\}. \tag{18}$$

## 2.2. Brownian dynamics

We can also generalize the above formalism to a non-unitary Brownian SYK model in which $J^{x,x+1}$ and/or $V^x$ are/is independent Gaussian random variables in time, with vanishing mean and the following variance

$$\overline{J_{i_1\ldots i_{p/2}j_1\ldots j_{p/2}}^{x,x+1}(\tau) J_{i_1\ldots i_{p/2}j_1\ldots j_{p/2}}^{x,x+1}(\tau')} = \delta(\tau - \tau') \frac{(p/2)!(p/2-1)!}{2JN^{p-1}}, \qquad \overline{V_{i_1\ldots i_q}^x(\tau) V_{i_1\ldots i_q}^x(\tau')} = \delta(\tau - \tau') \frac{(q-1)!}{VN^{q-1}}, \tag{19}$$

i.e. the correlation is non-vanishing only at equal-time. A variant of this model was studied in Ref. [44]. Previously, there has been extensive study on the Brownian unitary circuit models [44–50]. These models can provide analytical solutions for the quantum dynamics even with a small onsite Hilbert space and can sometimes show different dynamics from systems without randomness [51]. Here we plan to explore the influence of the temporal correlation on the non-unitary quantum dynamics.

One can apply the entire procedure of the last subsection to computing the Rényi entropy of the Brownian SYK model with minimal change. Note that in the Brownian model, a time-dependent random variable admits only equal-time correlation $t = t'$, which happens at the redefined times $s' = 2jt + s$ and $s' = 2(j+1)t - s$ on the contour for $Z_n(t)$. Therefore we need to introduce an additional projection to the Brownian term

$$P_B(s, s') = \sum_{j=1}^{2n} \delta(s - s' - 2jt) + \delta(s + s' - 2jt). \tag{20}$$

As an example, when the inter-site interaction becomes Brownian, the saddle point equation (16b) becomes

$$\Sigma_x^J(s,s') = \frac{J}{2} f(s) f(s') G_x^{\frac{p}{2}-1}(s,s') \left( G_{x+1}^{\frac{p}{2}}(s,s') + G_{x-1}^{\frac{p}{2}}(s,s') \right) P(s,s') P_B(s,s'). \tag{21a}$$

Similar modification applies when $V^x$ becomes Brownian. The on-shell action can still be computed as in (18).

## 2.3. Numerical details

The complex form of the saddle point equations (16) hampers analytical treatment for generic parameters of $J$, $V$, $t$, and $L_A$, and numerical methods are exploited instead. We will hereby focus on the second Rényi entropy with $n = 2$. We discretize $s \in [0, 8t]$ to 4L points, and represent $G_x$ and $\Sigma_x$ as $4L \times 4L$ matrices. Numerically, for moderate $J$ and $V$, we take $L = O(1) \times t$ and finite size scaling for L is performed as the final procedure to obtain the $L \to \infty$ result.

One solves the numerical version of Eqs. (16) iteratively: at the $n$th step, one plugs the Green's function obtained from the previous step, $G_x^{(n-1)}$, into the numerical form of Eqs. (16b) and (16c), then use Eq. (16a) to generate new Green's function to be used at the $(n+1)$th step, $G_x^{(n)}$. The saddle point solution is found when the Green's function series $\{G_x^{(n)}\}$ converge within numerical precision. The numerical version of (16) is

$$(G_x)_{ij} = \left[ (G_x^0)^{-1} - \Sigma_x^J - \Sigma_x^V \right]_{ij}^{-1}, \tag{22a}$$

$$\left( \Sigma_x^J \right)_{ij} = \frac{J^2}{2} f_i f_j (G_x)_{ij}^{\frac{p}{2}-1} \left( (G_{x+1})_{ij}^{\frac{p}{2}} + (G_{x-1})_{ij}^{\frac{p}{2}} \right) P_{ij}(\Delta t)^2, \tag{22b}$$

$$\left( \Sigma_x^V \right)_{ij} = V^2 (G_x)_{ij}^{q-1} P_{ij}(\Delta t)^2, \tag{22c}$$

where $\Delta t = 2t/L$. $P$ and $f$ are the straight-forward discretization of their definitions (see Eq. (15) and the paragraph below Eq. (8)):

$$P \quad = \quad \begin{array}{c}\text{[checkerboard matrix image]}\end{array}, \qquad f_j = \begin{cases} i, & j \in \left(\frac{L}{2}, L\right) \cup \left(\frac{3L}{2}, 2L\right) \cup \left(\frac{5L}{2}, 3L\right) \cup \left(\frac{7L}{2}, 4L\right), \\ -i, & \text{otherwise.} \end{cases}$$

where we have pictorially presented the $4L \times 4L$ matrix of $P$, with the gray (white) representing unit (vanishing) entries . The partial operator $\partial_\tau$ in the continuum version (16) is now discretized in Eqs. (22) to the non-interacting Green's function $(G_x^0)^{-1}$

$$\begin{aligned} x \in A: \quad &\left( G_x^0 \right)_{ij} \equiv \left( G_A^0 \right)_{ij} = \frac{1}{2} \text{sgn}(i-j) \quad \text{for } i, j \in \{1, 2, ..., 2L\} \text{ or } \in \{2L+1, ..., 4L\}, \\ x \in B: \quad &\left( G_x^0 \right)_{ij} \equiv \left( G_B^0 \right)_{ij} = \left( G_A^0 \right)_{i-L, j-L}, \end{aligned} \tag{23}$$

where $\text{sgn}(x)$ is the sign function with $\text{sgn}(0) = 0$, the row and column indices are understood in the modulo sense $i + 4L \equiv i$, and the unmentioned entries are identically zero. Note this substitution $\partial_\tau \to (G_x^0)^{-1}$ is crucial to ensure numerically accurate result with small number of points. We introduce the following terminology: a solution is *replica diagonal* if for all $x$, the Green's function $G_x^0$ is non-zero only for elements that are non-zero in $G_x^{0\,2}$, and a solution is *replica quasi-diagonal* if this property is satisfied for all $x$ in the subsystem bulks (i.e. interior) $\mathring{A}$, $\mathring{B}$ and is violated on the subsystem boundaries $\partial A$ and $\partial B$.

We consider two different types of initial conditions:

Type 1(A). $\quad G_x^{(0)} = G_A^0 \quad$ for $x \in [1, L]$,

Type 1(B). $\quad G_x^{(0)} = G_B^0 \quad$ for $x \in [1, L]$,

Type 2(D). $\quad G_x^{(0)} = G_A^0 \quad$ for $x \in A \quad$ and $\quad G_x^{(0)} = G_B^0 \quad$ for $x \in B$, $\tag{24}$

Type 2(QA). $\quad G_x^{(0)} = G_A^0 \quad$ for $x \in \mathring{A}$, $\quad G_x^{(0)} = G_B^0 \quad$ for $x \in B$ and $\quad G_x^{(0)} = 0.9 G_A^0 + 0.1 G_B^0 \quad$ for $x \in \partial A$,

Type 2(QB). $\quad G_x^{(0)} = G_A^0 \quad$ for $x \in A$, $\quad G_x^{(0)} = G_B^0 \quad$ for $x \in \mathring{B}$ and $\quad G_x^{(0)} = 0.9 G_B^0 + 0.1 G_A^0 \quad$ for $x \in \partial B$,

---

[2] The replica diagonal solution here refers to diagonality with respect to the Rényi index $n$. This is different from the diagonality in the disorder replica space mentioned in Subsection 2.1.

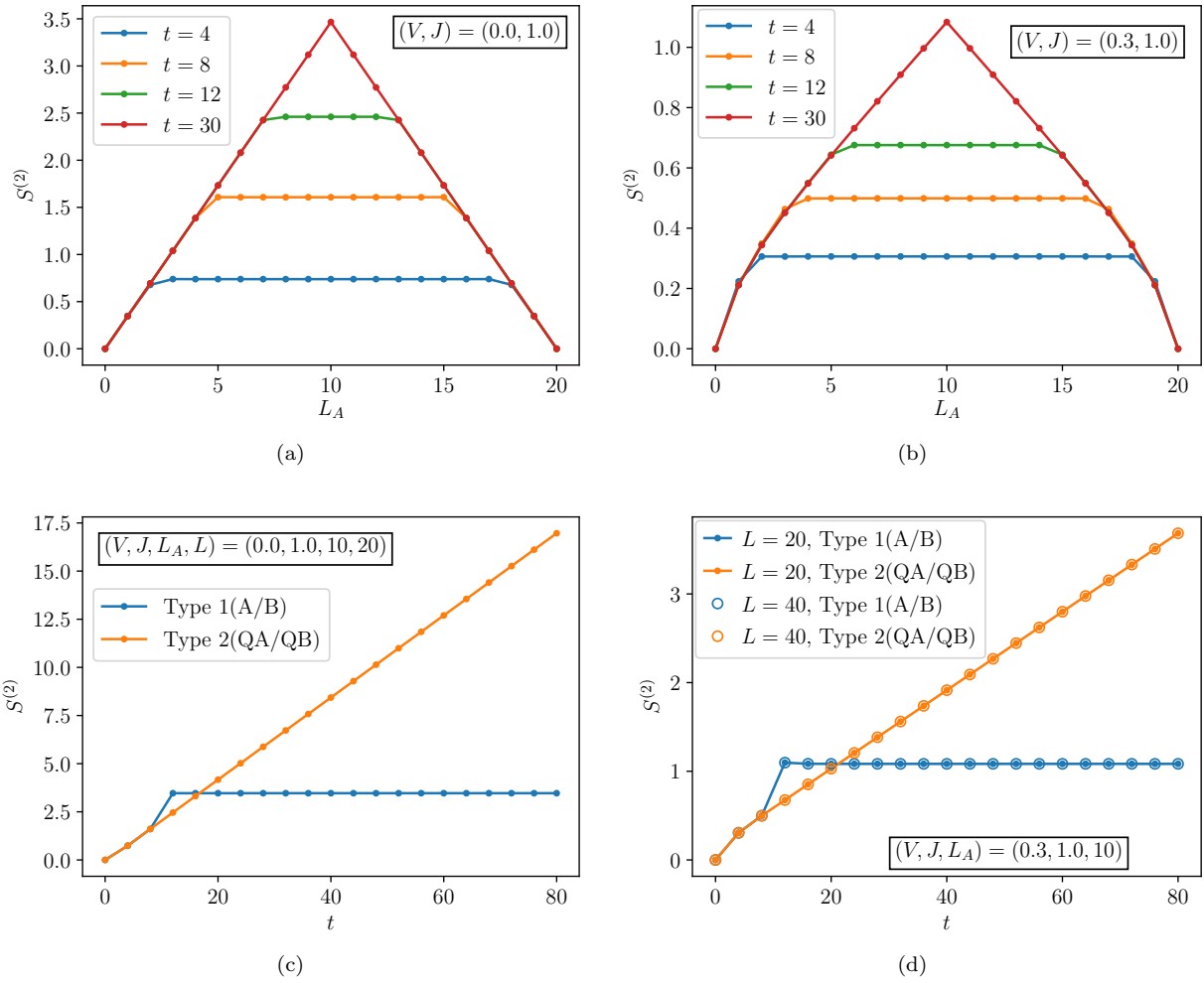

Figure 3. (a) Subsystem scaling of $S^{(2)}$ at various $t$ with $V = 0$. (b) Subsystem scaling of $S^{(2)}$ at various $t$ with $V = 0.3$. (c) Time dependence of the two saddle solutions at $V = 0$, obtained by using numerical initial conditions Type 1(A/B) and Type 2 (QA/QB), respectively. (d) Time dependence of the two saddle point solutions at $V = 0.3$. The result from two system sizes $L = 20$ and $L = 40$ shows there is no observable finite size effect.

By definition, the initial condition Type 2(D) is replica diagonal, and the initial condition Type 2(QA/QB) is replica quasi-diagonal. When $q \geq 4$, the replica diagonal/quasi-diagonal property is preserved under the iteration and the initial condition Type 2(D) (Type 2(QA/QB)) always leads to replica diagonal (quasi-diagonal) self-consistent solution. We have also checked that other initial conditions would not lead to new saddle points.

After solving Eqs. (22), we compute $S_{2,\text{saddle}}(L_A)$ as a function of the subsystem size $L_A$ using the discretized version of Eq. (18). The final result is obtained by using Eq. (17)

$$S^{(2)}/N = S_{n=2,\text{saddle}}(L_A) - S_{n=2,\text{saddle}}(0). \tag{25}$$

## 3. PHYSICAL PROPERTIES OF THE MODEL

In this section we present numerical results along with analytical understandings for our model with different $(p, q)$. We focus on models with $(p, q) = (4, 2)$ and $(2, 2)$ in the next two subsections. For the $(4, 2)$ model, we find that the steady state is a volume-law entanglement state for $V \ll J$ and is an area-law entanglement state for $J \ll V$. These two phases are separated by a first order transition. On the other hand, for the $(2, 2)$ model, we find the steady state is always in a critical phase for finite $V/J$. We will briefly mention properties of the $(p, q) = (4, 4)$ model in Appendix C.

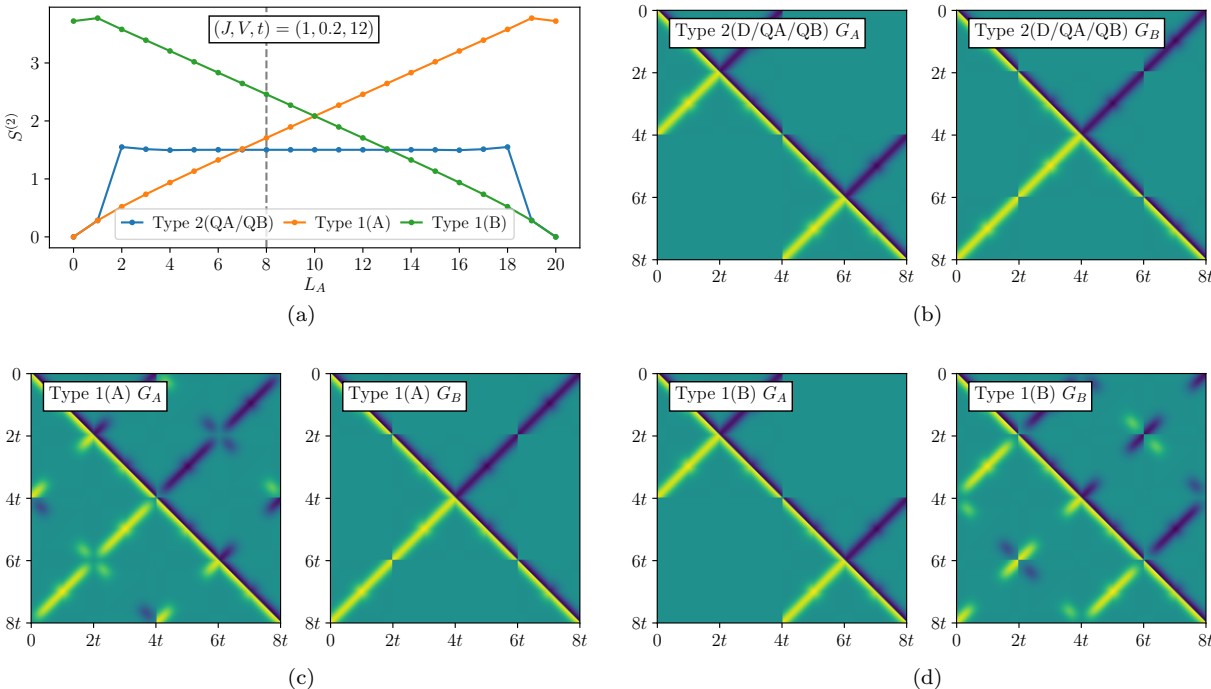

Figure 4. (a) Subsystem scaling of $S^{(2)}$ obtained from different saddle point solutions: Type 2(QA/QB) saddle, Type 1(A) saddle, and Type 1(B) saddle. (b)–(d) The Green's function profile in the subsystem bulks $\mathring{A}$ and $\mathring{B}$ for $L_A = 8$ (the dashed line in (a)) corresponding to (b) the Type 2(D/QA/QB) saddles, (c) the Type 1(A) saddle, and (d) the Type 1(B) saddle.

### 3.1. Inter-cluster SYK$_4$ with intra-cluster SYK$_2$

The $(p, q) = (4, 2)$ model consists of Hermitian, inter-cluster SYK$_4$ interaction with strength $J$ and non-Hermitian, intra-cluster SYK$_2$ interaction with strength $V$. We prepare a product state and let it evolve under this Hamiltonian described by Eq. (6). We study the scaling of the second Rényi entropy $S^{(2)}$ of the state as a function of $J$, $V$, subsystem size $L_A$, and evolution time $t$.

#### 3.1.1. $V \ll J$

Before discussing the non-unitary dynamics, we first consider the unitary dynamics at $V = 0$. Since the system has local interaction, we expect the entanglement entropy for a subsystem grows linearly in time and saturates to a constant proportional to $L_A$ at late time. We compute $S^{(2)}$ by numerically solving the saddle point solution and present the results in in Fig. 3(a). In summary, the entanglement dynamics satisfies

$$S^{(2)}(t, L_A) = \begin{cases} v_E t & \text{if} \quad t < s_0 L_A / v_E \\ s_0 L_A & \text{if} \quad t \geq s_0 L_A / v_E \end{cases} \tag{26}$$

where the entanglement velocity $v_E$ depends on the magnitude of $J$ and the transition occurs at $t = s_0 L_A / v_E$. Since the initial state we consider has energy expectation value $\langle E \rangle = 0$, the final steady state is indeed a maximally entangled thermal state at infinite temperature with $s_0 = \log 2/2$. Notice that the entanglement dynamics in the large $N$ model is distinct from that in a chaotic system with finite or small $N$, in which the energy conservation law leads to a diffusive dynamics for $S^{(2)}(t)$, i.e., $S^{(2)}(t) \sim \sqrt{t}$ [52, 53]. This slow dynamics disappears in the SYK chain model in the large $N$ limit.

The above behavior still holds when $V$ takes a small finite value, although with $s_0 < \log 2/2$. For instance, we present the $V = 0.3$ result in Fig. 3(b), which is analogous to the $V = 0$ result. This indicates that when $V \ll J$, there exists a phase in which the steady state entanglement entropy has volume law scaling.

Here we provide an explain for this entanglement scaling from the saddle point solution perspective. The $S^{(2)}$ in Fig. 3(a) and Fig. 3(b) is determined by the minimum value of the various saddle point solutions which are obtained by

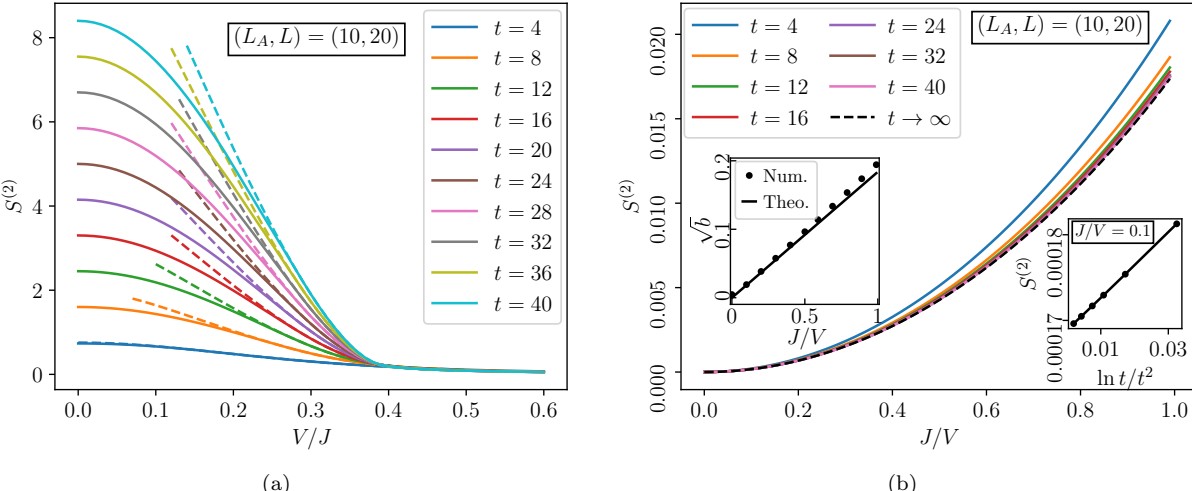

Figure 5. (a) The $S^{(2)}$ obtained from Type 2(D/QA/QB) saddles as a function of $V/J$ for $V < 0.6J$ at various times. For a given value of $V$ and $t$, the solid and dashed lines, if both exist, correspond to the Type 2(D) (i.e. replica diagonal) and Type 2(QA/QB) (i.e. replica quasi-diagonal) saddle, respectively; otherwise only Type 2(D) saddle solution exists for this value of $V$. Note the Type 1 saddle solutions are not shown in (a). (b) The Type 2(D) saddle $S^{(2)}$ as a function of $J/V$ for $J \leq V$. Long time behavior converges to a $J^2/V^2$ scaling form. The left inset shows that the coefficient $b$ in front of a $\log t/t^2$ correction term fits well with the theoretically proposed value (see Eq. (30) and below); the right inset shows the linear $S^{(2)} - \log t/t^2$ scaling behavior at small $J/V$.

choosing the initial conditions Type 1(A/B) and Type 2(D/QA/QB) as mentioned in Sec. 2.3. As shown in Fig. 4(a), starting from an initial condition of Type 1, $S^{(2)}$ quickly saturates to a volume law scaling, including a branch with $S^{(2)} \propto L_A$ (Type 1(A)) and a branch with $S^{(2)} \propto L_B$ (Type 1(B)). On the other hand, initial condition Type 2 always leads to a constant $S^{(2)}$ that is insensitive to subsystem size. Furthermore, for small $V \lesssim 0.3$, we observe that the replica quasi-diagonal saddle obtained using Type 2(QA/QB) and the replica diagonal saddle obtained using Type 2(D) give slightly different results. Both solutions give $S^{(2)}$ that grows linearly in time, however the replica quasi-diagonal saddle has a lower $S^{(2)}$ value (see Fig. 5(a)). Interestingly, as $V$ increases, the replica quasi-diagonal saddle gradually merges with the replica diagonal saddle, and as $V > 0.3$, both initial conditions Type 2(Q) and Type 2(D) give the same replica diagonal solution. We will henceforth refer to the saddle point solutions by the initial conditions that lead to them.

The above analysis suggests that early time $S^{(2)}$ is determined by the Type 2 saddle which grows linearly in time, and that late time $S^{(2)}$ is determined by the Type 1 saddle which eventually saturates to a constant that is proportional to $L_A$. The time dynamics is presented in Fig. 3(c) and Fig. 3(d) for $V = 0$ and $V = 0.3$ respectively and is consistent with the result in Eq. (26). In addition, in Fig. 3(d), we verify that no finite size effect exists in both saddle solutions.

We can further understand the Type 1 and Type 2 saddles by analyzing the Green's functions. Here we take $V = 0.2$ as an example. The $S^{(2)}$ computed from different saddles are plotted in Fig. 4(a), and the subsystem bulk Green's functions for different saddles are shown in Figs. 4(b)–4(d). The Green's function corresponding to Type 2 saddles has a replica diagonal form (Fig. 4(b)) inside the subsystem bulk. This means there exists self-consistent replica diagonal/quasi-diagonal solution of the Green's functions when computing $Z_n(t)$. Compared to the Green's functions used for computing the normalization $Z(t)$ (with proper time labeling), the Green's functions for $Z_n(t)$ are modified only in the neighborhood of subsystem boundaries, since only near the boundaries is there a change of the Schwinger-Dyson equation. Consequently, terms in $Z_n(t)$ and $nZ(t)$ that involve only Green's functions in the subsystem bulk $\mathring{A}$ and $\mathring{B}$ cancel, and the Rényi entropy receives contribution only from the Green's functions on the subsystem boundaries $\partial A$ and $\partial B$. This leads to a $S^{(2)}$ which is independent of the subsystem size. On the contrary, if we take a replica non-diagonal saddle as in Fig. 4(c) and Fig. 4(d), the Green's functions become highly replica non-diagonal in one of the subsystems ($A$ or $B$, depending on the initial conditions being Type 1(A) or Type 1(B)). Consequently, there is non-vanishing contribution to the entropy even deeply inside the subsystem bulk. This may give rise to a $S^{(2)}$ proportional to the subsystem size. We mention in passing that the Green's functions also distinguish replica diagonal and replica quasi-diagonal saddles due to their difference at the subsystem boundaries. As an example, we show in Fig. 6 the Green's functions of the replica quasi-diagonal saddle on subsystem boundaries for various $V$. The corresponding Green's functions in the subsystem bulk (not plotted) all have the same profile as

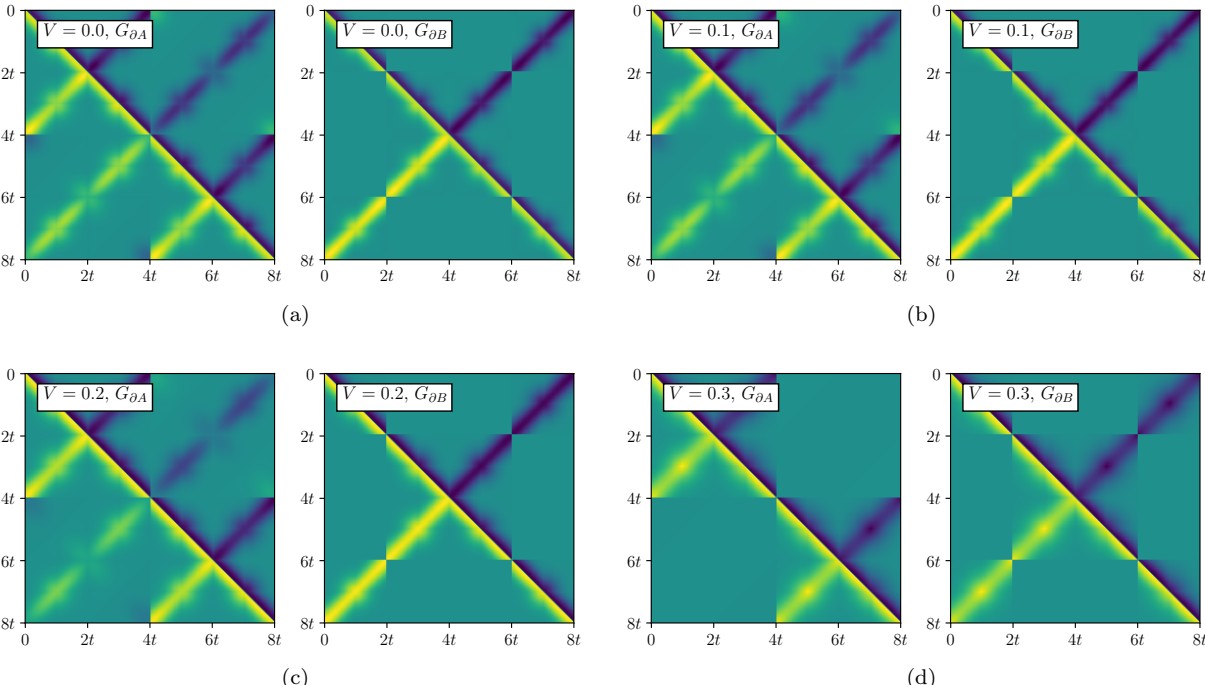

Figure 6. Green's function profile on the subsystem boundary $\partial A$ and $\partial B$ for various $V$ at $t = 8$. Initial condition Type 2(QA) is applied. As $V$ increases, the saddle point solution gradually changes from a replica quasi-diagonal solution to a replica diagonal solution.

in Fig. 4(b). The boundary Green's function profiles confirm that as $V$ increases, the replica quasi-diagonal solution converges to the replica diagonal solution.

Interestingly, the simultaneous existence of such replica non-diagonal and replica diagonal/quasi-diagonal solutions is directly related to the recent resolution of the black hole information paradox. In the language of gravity theories, such highly replica non-diagonal solution is known as a "replica wormhole" [38, 39]. In the story of the information paradox, without such replica wormhole, the entropy of the system would grow unboundedly, which ultimately violates unitarity. We find similar behavior exists in our model: without finding the Type 1 saddles, we would conclude that the entropy of our system grows linearly, finally exceeding the maximal entropy $L_A \log 2/2$. When the "replica wormhole" saddle point is taken into account, the entropy becomes well-behaved.

### 3.1.2. $V \gg J$

Now we turn to the other limit $J/V \to 0$. First, the $J = 0$ case defines purely imaginary evolution governed by the intra-cluster $\mathrm{SYK}_2$ interaction. The system evolves into its ground state after long time evolution. Since the $\mathrm{SYK}_2$ model is a free fermion model and has vanishing zero temperature entropy [20], the ground state is a simple product state, i.e.,

$$|\Psi_{\mathrm{GS}}\rangle = |\psi_{\mathrm{GS}}^1\rangle|\psi_{\mathrm{GS}}^2\rangle \cdots |\psi_{\mathrm{GS}}^L\rangle \tag{27}$$

where $|\psi_{\mathrm{GS}}^x\rangle$ denotes the $\mathrm{SYK}_2$ ground state at site $x$ and there is no entanglement between different sites. When $J \neq 0$, we expect that the steady state entropy $S^{(2)}$ still satisfies area law and confirm this result in Fig. 7(a). This scaling behavior can be understood as follows: As long as $J \ll V$, we can perturbatively study $S^{(2)}$ by taking the Type 2(D) saddle solution. The leading behavior of $S^{(2)}$ receives corrections only from the subsystem boundaries and can be written in closed form (see Appendix B for derivation):

$$S^{(2)} = \delta S = \frac{4J^2}{p} \int_0^t ds \int_t^{2t} ds' f(s) f(s') G_c^p \tag{28}$$

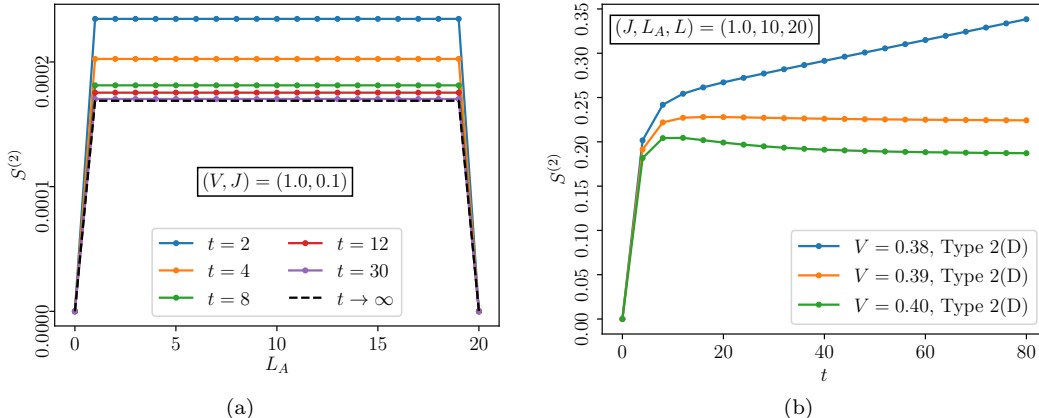

Figure 7. (a) The result for $S^{(2)}$ vs $L_A$ at different times at $J = 0.1$. (b) The Type 2 saddle around the transition point.

where $G_c$ is the conformally invariant Green's function

$$G_c(s, s') = \frac{1}{2tV \sin \frac{\pi(s-s')}{2t}},\tag{29}$$

for the ground state of SYK$_2$ valid for $s - s' \gg \epsilon \approx V^{-1}$[3]. Plugging $G_c$ into Eq. (28), we get

$$S^{(2)} = a\frac{J^2}{V^2} + b\frac{\log(Vt)}{V^2t^2} + c\frac{J^2}{V^4t^2} + \frac{1}{t^2}O\left((\epsilon/t)^2\right), \quad t \gg V^{-1},\tag{30}$$

where the prefactor of the logarithmic term $b = \frac{J^2}{3\pi^2V^2}$ is cutoff-independent, while $a$, $c$ are cutoff-dependent finite constant. The first term of $S^{(2)}$ is responsible for the area law scaling behavior of the steady state and the $J^2/V^2$ scaling form is confirmed numerically in Fig. 5(b). The second term contains the leading time dependence at small $J/V$ which shows a decaying behavior that is linear in $\log t/t^2$, see the right inset of Fig. 5(b). We also verify the explicit form of the coefficient $b$, see the left inset of Fig. 5(b).

### 3.1.3. The transition at the intermediate regime

The above analysis suggests that there exists a volume law phase when $V \ll J$ and an area law phase when $V \gg J$. We further explore the intermediate regime and find that there is a phase transition separating these two phases. In Fig. 7, we analyze the Type 2 saddle solution for $S^{(2)}$ as a function of time and we find that it changes dramatically across $V/J = 0.39$. When $V/J = 0.38$, it grows linearly in time while when $V/J = 0.39$ or 0.4, it decays with time and saturates to a constant. This result indicates that the transition is first order and occurs close to 0.39. In the volume law phase, the steady state entanglement entropy is determined by the Type 1 saddle solution, which scales linearly with the subsystem size up to $L/2$. The slope decreases as we increase $V/J$. As we enter the area law phase, $S^{(2)}$ is bounded by the Type 2 saddle solution which decays with time and saturates to a finite constant.

We further analyze the volume law phase close to the transition point, $V/J \lesssim 0.39$, and find that the behavior of $S^{(2)}$ is slightly different from that of the $V \leq 0.3$ region at early time. We take $V/J = 0.36$ as an example, see Fig. 8(a). When the subsystem size is small, $S^{(2)} \sim \alpha L_A$, with the coefficient $\alpha = \alpha(t)$ a decreasing function in time. This decaying behavior originates from the Type 1 saddle solution and is responsible for the spike observed in Fig. 8(b). At late time, this coefficient saturates to a finite constant and the steady state still exhibits volume law scaling.

We summarize the main result for the $(4, 2)$ model in the phase diagram shown in Fig. 9. Three distinct early time behaviors of $S^{(2)}(L_A)$ (dashed and dotted lines) as well as two distinct late time behaviors (red lines) are sketched.

---

[3] This means the outer integral of Eq. (28) has to be regulated as $\int_\epsilon^{t-\epsilon} ds$.

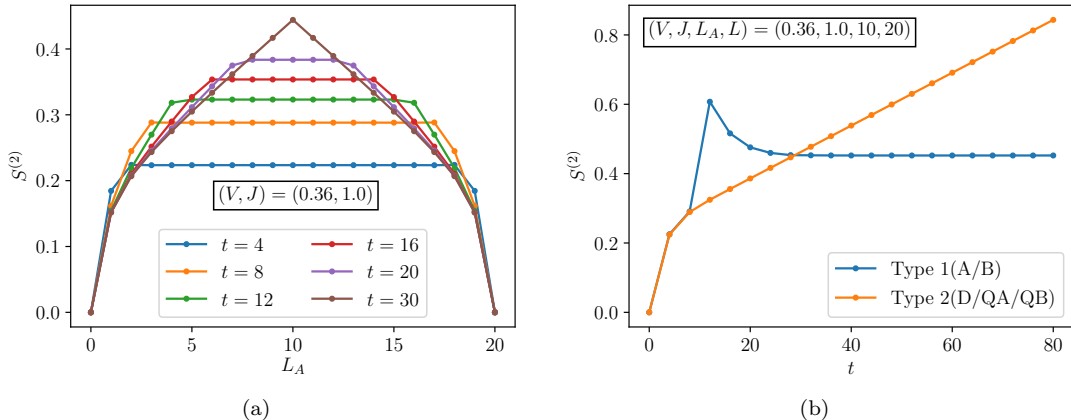

Figure 8. (a) The result for $S^{(2)}$ vs $L_A$ at different times at $V = 0.36$. (b) The comparison between two saddles.

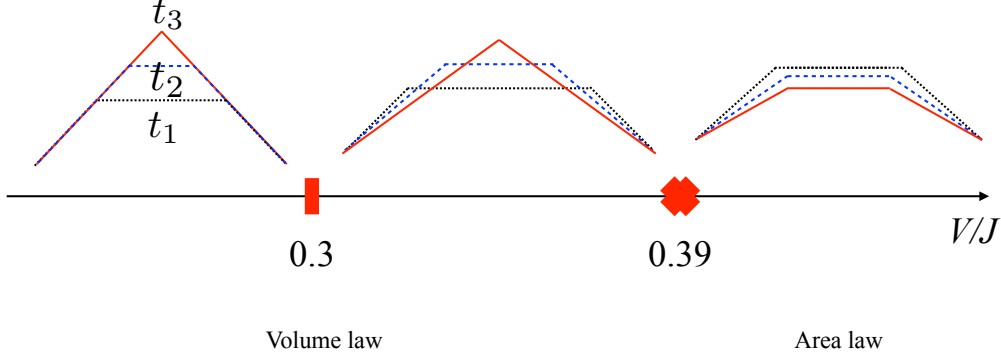

Figure 9. The schematic phase diagram for the $(4, 2)$ model. Here $t_1 < t_2 < t_3$. The phase transition occurs at $V/J = 0.39$.

Early time entropy always contains a plateau, corresponding to a Type 2 (replica diagonal/quasi-diagonal) solution. The late time, steady state entropy defines either a volume law phase or an area law phase, corresponding to a Type 1 (replica non-diagonal) solution or a Type 2 solution, respectively.

### 3.1.4. Brownian version

By assuming equal-time (and otherwise vanishing) correlation for the disorder variables $J$ and/or $V$ (see the definition in Eq.(19)), a Brownian $(p, q) = (4, 2)$ model can be written down and studied analogously. Following the general recipe of Subsection 2.2, we consider two Brownian versions: 1) the $(4_\mathrm{B}, 2)$ model, in which only the inter-cluster SYK$_4$ interaction $J$ becomes Brownian, and 2) the $(4_\mathrm{B}, 2_\mathrm{B})$ model, in which both $J$ and the intra-cluster SYK$_2$ interaction $V$ are Brownian.

We find that the $(4_\mathrm{B}, 2)$ model is in many ways similar to the $(4, 2)$ model: the steady state exhibits volume-law entanglement for $V \ll J$ and area-law entanglement for $V \gg J$. The two phases are separated by a first order phase transition. We further compare the $(4_\mathrm{B}, 2)$ model and the $(4, 2)$ model as follows:

- In the $V \ll J$ regime, both models satisfy the early and late time dynamics (26). The $J = 0$ limit has the same coefficient $s_0 = \log 2/2$. The $(4_\mathrm{B}, 2)$ model has a much smaller coefficient $v_E$.

- The initial conditions Type 2(D) and Type 2(QA/QB) converge to distinct saddle point solutions in the $(4, 2)$ model when $V < 0.3J$. They converge to the same solution in the $(4_\mathrm{B}, 2)$ model, which is replica diagonal for all $V/J$.

- In the $V \gg J$ regime, the steady state of the $(4_\mathrm{B}, 2)$ model has the scaling form $S^{(2)} \propto J/V$. This is different from the $S^{(2)} \propto J^2/V^2$ scaling of the $(4, 2)$ model and is attributed to the presence of $J^{-1}$ in the Brownian

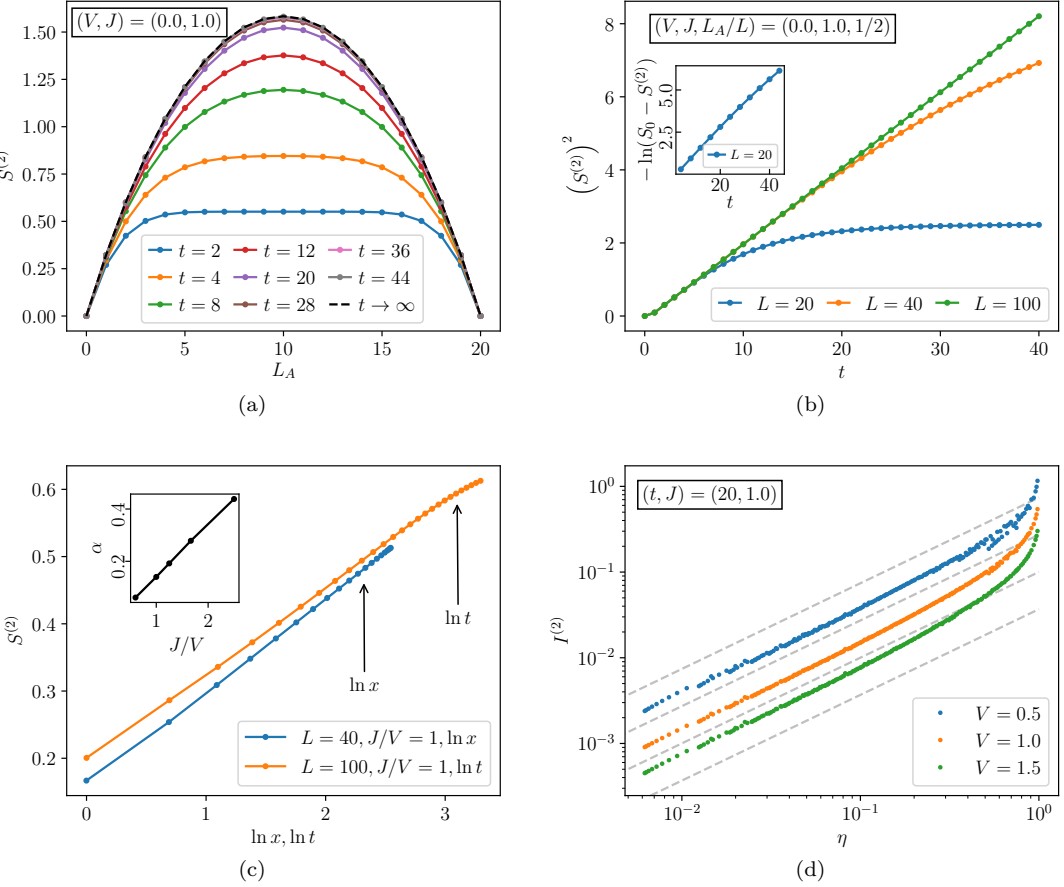

Figure 10. (a) Subsystem scaling of $S^{(2)}$ at $V = 0$. The $t \to \infty$ curve is plotted from the analytical result Eqs. (31)–(32). (b) $S^{(2)}$ as a function of time at $V = 0$. An early time scaling $S^{(2)} \propto \sqrt{t}$ can be observed in large enough systems. The inset shows that late time $S^{(2)}$ exponentially saturates to $S_0 = \frac{L}{4}(3\log 2 - 2\log(\sqrt{2}+1))$. (c) Subsystem scaling of $S^{(2)}$ in the intermediate regime $V/J \sim 1$. $S^{(2)}$ scales linearly with both $\log t$ and $\log x$ ( $x \equiv \log[L\sin(\pi L_A/L)/\pi]$) with very close slopes $\alpha$. The inset shows $\alpha$ scales linearly with $J/V$. (d) Mutual information $I^{(2)}$ as a function of subsystem cross-ratio $\eta$ at various $V/J$.

correlation relation (19).

• The first order transition occurs at $V/J \sim 0.07$ for the $(4_\mathrm{B}, 2)$ model, while it occurs at $V/J \simeq 0.39$ for the $(4, 2)$ model.

Unlike the $(4_\mathrm{B}, 2)$ model, the $(4_\mathrm{B}, 2_\mathrm{B})$ model does not exhibit phase transitions. The entanglement dynamics in the entire parameter range of $V/J$ is described by Eq. (26), and the steady state is a volume-law entanglement state [4].

### 3.2. Inter-cluster SYK$_2$ with intra-cluster SYK$_2$

Another variant of the model is the $(p, q) = (2, 2)$ model. This free fermion model exhibits critical phases distinct from the previous interacting model. We again first consider the limit $V = 0$. The entropy $S^{(2)}$ as a function of subsystem size and time is shown in Figs. 10(a) and 10(b). We observe that $S^{(2)}$ at early time increases linearly with $\sqrt{t}$, and at late time exponentially saturates to a steady state value (see the inset of Fig. 10(b)). The subsystem scaling of this steady state value agrees well with the following expression (see Fig. 10(a))

$$S^{(2)} = L S^{(2)}_{\mathrm{SYK2}}(L_A/L), \tag{31}$$

---

[4] In this case, the replica diagonal saddle point can be solved analytically and its entropy is always linear in $t$.

where $S^{(2)}_{\text{SYK2}}(\lambda)$ is the eigenstate Rényi entanglement entropy for a subsystem with $N\lambda$ Majorana modes of a single-cluster SYK$_2$ model with $N$ Majorana modes [31, 34, 54, 55]:

$$S^{(2)}_{\text{SYK2}}(\lambda) = \frac{3}{2}\lambda\log 2 + \left(\frac{1}{2} - \lambda\right)\log\left(\frac{\sqrt{1 + 4\lambda - 4\lambda^2} + 1 - 2\lambda}{1 - \lambda}\right) - \frac{1}{2}\log\left(\sqrt{1 + 4\lambda - 4\lambda^2} + 1\right). \tag{32}$$

This result is much smaller than the $(\log 2/2)L_A$ scaling in the $(4,2)$ model with $V = 0$.

The entanglement entropy takes a different form when $V \neq 0$. When $V = J = 1$, as shown in Fig. 10(c), the steady state entropy has the form

$$S^{(2)} = \alpha\log\left[\frac{L}{\pi}\sin(\frac{\pi L_A}{L})\right], \tag{33}$$

which is the same as that for the ground state of a 1D critical system with periodic boundary condition. Furthermore, starting from a product state, $S^{(2)}$ increases as

$$S^{(2)} = \alpha\log t \tag{34}$$

at early time with the same coefficient $\alpha$. Such scaling behavior implies an emergent two-dimensional conformal symmetry with dynamical exponent $z = 1$. Similar behavior has also been found in the random non-unitary free fermion dynamics in the small $N$ limit [56].

We then tune away from the ratio $V/J = 1$ and find that the critical scaling behavior of $S^{(2)}$ is retained in a large parameter range. More interestingly, we observe that the coefficient $\alpha$ is linearly proportional to the ratio $J/V$ (see inset of Fig. 10(c)), indicating that this model remains critical as long as $V/J$ is finite.

In addition, we compute the second Rényi mutual information $I^{(2)}$ of the steady state defined as

$$I^{(2)} = S^{(2)}_{[x_1, x_2]} + S^{(2)}_{[x_3, x_4]} - S^{(2)}_{[x_1, x_2]\cup[x_3, x_4]}. \tag{35}$$

Here we take periodic boundary condition and partition the system into four connected intervals with end points $x_{1,2,3,4}$. We present $I^{(2)}$ in Fig. 10(d) and find that it is only a function of cross-ratio, which is defined as

$$\eta = \frac{\sin\frac{\pi}{L}|x_1 - x_2|\sin\frac{\pi}{L}|x_3 - x_4|}{\sin\frac{\pi}{L}|x_1 - x_3|\sin\frac{\pi}{L}|x_2 - x_4|}. \tag{36}$$

This numerical result provides further evidence that this model has conformal symmetry. In particular, we find that when $\eta \to 0$, $I^{(2)} \sim \eta$, the same as that for non-unitary random free fermion dynamics in the small $N$ limit [56].

The very short time behavior of $S^{(2)}$ in the limit $J \ll V$ can be treated perturbatively, see Appendix C.

## 4. DISCUSSION AND CONCLUSION

In this work we study from the entanglement perspective the non-unitary dynamics of a 1D SYK chain $H = J\sum_x H^p_{x,x+1} - iV\sum_x H^q_x$. We derive the large $N$ self-consistent saddle point solution to the $n$th Rényi entropy using the path integral method, and numerically study the second Rényi entropy as a function of time, subsystem size, and the dimensionless coupling strength. We find that for the $(p,q) = (4,2)$ model, as we vary the ratio $V/J$, the steady state can be either in a volume-law phase or an area-law phase. The steady states of these two phases correspond to distinct saddle point solutions, and the two phases are separated by a first order transition. For the non-interacting $(p,q) = (2,2)$ model, the steady state exhibits critical behavior for any finite ratio $V/J$, indicating the emergent two-dimensional conformal symmetry in this model.

There are a lot of possible interesting extensions of our work, which we briefly mention. Firstly, in the $(4,2)$ model, the transition is first order when inter-cluster coupling is described by the regular SYK or Brownian SYK interaction. It would be interesting to study the $1/N$ correction in this model and check if it can round this first order transition to a second order transition. Secondly, the emergent conformal symmetry observed in $(2,2)$ model may need further investigation. We plan to analytically study the critical saddle point in this large $N$ model and find the connection with the critical phenomena observed in the small $N$ model. Lastly, we can use the method developed in this paper to construct other non-unitary dynamics and explore the exotic phases in them.

*Acknowledgment* We acknowledge helpful discussions with Leon Balents, Yaodong Li and Andy Lucas. PZ acknowledges support from the Walter Burke Institute for Theoretical Physics at Caltech. CL is supported by the NSF CMMT program under Grants No. DMR-1818533. Use was made of computational facilities purchased with funds from the National Science Foundation (CNS-1725797) and administered by the Center for Scientific Computing (CSC). The CSC is supported by the California NanoSystems Institute and the Materials Research Science and Engineering Center (MRSEC; NSF DMR-1720256) at UC Santa Barbara.

## Appendix A: Derivation of Eq. (14)

In this section we provide the detailed derivation for the bilocal field action (14). By definition, we have

$$
\begin{aligned}
Z_n[\{J_{i_1\ldots i_{p/2}j_1\ldots j_{p/2}}^{x,x+1}, V_{i_1\ldots i_q}^x\}](t, L_A) = \mathrm{Tr}_A \left( \underbrace{\mathrm{Tr}_B \left[ e^{-itH}|\{0\}\rangle\langle\{0\}|e^{itH^\dagger} \right] \cdots \mathrm{Tr}_B \left[ e^{-itH}|\{0\}\rangle\langle\{0\}|e^{itH^\dagger} \right]}_{n \text{ copies of } \mathrm{Tr}_B} \right) \\
= \sum_{\phi_A} \langle\phi_A| \sum_{\phi_B^1} \langle\phi_B^1|e^{-itJH^p-tVH^q}|\{0\}\rangle\langle\{0\}|e^{itJH^p-tVH^q}|\phi_B^1\rangle \cdots \\
\sum_{\phi_B^n} \langle\phi_B^n|e^{-itJH^p-tVH^q}|\{0\}\rangle\langle\{0\}|e^{itJH^p-tVH^q}|\phi_B^n\rangle |\phi_A\rangle,
\end{aligned}
\tag{A1}
$$

where $1_{A/B} = \sum_{\phi_{A/B}} |\phi_{A/B}\rangle\langle\phi_{A/B}|$ is the completeness relation for the Hilbert subspaces $\mathcal{H}_A$ or $\mathcal{H}_B$. Note that when using coherent states, additional minus signs lead to the anti-periodic boundary condition of fermions as for thermal ensembles. We now conduct disorder average over the disorder fields $J_{i_1\ldots i_{p/2}j_1\ldots j_{p/2}}^{x,x+1}$ and $V_{i_1 i_2\ldots i_q}^x$ and get

$$
\overline{Z_n} = \int \prod dJ_{i_1\ldots i_{p/2}j_1\ldots j_{p/2}}^{x,x+1} dV_{i_1\ldots i_q}^x P(J_{i_1\ldots i_{p/2}j_1\ldots j_{p/2}}^{x,x+1}, V_{i_1\ldots i_q}^x) Z_n[\{J_{i_1\ldots i_{p/2}j_1\ldots j_{p/2}}^{x,x+1}, V_{i_1\ldots i_q}^x\}] = \int D[\chi_i^x]e^{-S_n[\{\chi_i^x\}]},
\tag{A2}
$$

here $P(J_{i_1\ldots i_{p/2}j_1\ldots j_{p/2}}^{x,x+1}, V_{i_1\ldots i_q}^x)$ is the Gaussian distribution of random parameters and

$$
\begin{aligned}
S_n = \sum_x \Bigg\{ \sum_i \int_0^{2nt} \frac{1}{2}\chi_i^x \partial_\tau \chi_i^x - \\
\frac{1}{2}\int_{[0,2nt]^2} d\tau d\tau' \left[ f(\tau)f(\tau')\frac{J^2}{p}\left(\sum_i \chi_i^x(\tau)\chi_i^x(\tau')\right)^{\frac{p}{2}}\left(\sum_i \chi_i^{x+1}(\tau)\chi_i^{x+1}(\tau')\right)^{\frac{p}{2}} + \frac{V^2}{q}\left(\sum_i \chi_i^x(\tau)\chi_i^x(\tau')\right)^q \right] \Bigg\},
\end{aligned}
\tag{A3}
$$

the factor $f(\tau) = \pm i$ has been introduced in the main text (see below Eq. (8)). The boundary conditions for $\chi_i^x$ occur at times $\tau = 0, 2t, 4t, \ldots, 2nt$ due to the trace over $A$ or $B$ and at times $\tau = t, 3t, \ldots, (2n-1)t$ due to the existence of $|\{0\}\rangle\langle\{0\}|$. The latter conditions can be incorporated by redefinition of the Majorana fields:

$$
\widetilde{\chi}_i^x(s) = \begin{cases} \chi_{2j}^x(2mt + (s - 4mt)), & s \in [4mt, (4m+1)t] \\ -i\chi_{2j-1}^x((2m+1)t + (4m+1)t - s), & s \in [(4m+1)t, (4m+3)t] \\ -\chi_{2j}^x((2m-1)t + s - (4m+3)t), & s \in [(4m+3)t, (4m+4)t] \end{cases}
\tag{A4}
$$

for $m = 1, 2, \ldots, n$, with $n + 1 \equiv 1$. The path defined for $\widetilde{\chi}(s)$ on $s \in [0, 4nt)$ is shown in Fig. 2 (where the "~" is removed). We see that the only boundary conditions needed to specify for $\chi$ are

$$
\begin{aligned}
x \in A: & \quad \widetilde{\chi}_i^x(4mt^+) = -\widetilde{\chi}_i^x((4m+4)t^-), \\
x \in B: & \quad \widetilde{\chi}_i^x((4m+2)t^-) = \widetilde{\chi}_i^x((4m+6)t^+), \\
& \quad \widetilde{\chi}_i^x(4mt^+) = -\widetilde{\chi}_i^x((4m+8)t^-).
\end{aligned}
\tag{A5}
$$

Eqs. (A4) and (A5) together define the contour $\mathcal{C}$. Note that the partial operator will also inherit this boundary condition, which we denote as $\widetilde{\partial}_{s,x}$. On contour $\mathcal{C}$ the direction of real-time evolution has the explicit form

$$
f(s) = \begin{cases} i & \text{for } s \in [(2m+1)t, (2m+2)t), \\ -i & \text{for } s \in [2mt, (2m+1)t). \end{cases}
\tag{A6}
$$

Next we introduce bilocal fields

$$
1 = \int_{\mathcal{C}} [D\widetilde{G}_x][D\widetilde{\Sigma}_x]e^{-\frac{1}{2}\sum_x \int d\tau d\tau' \widetilde{\Sigma}_x(\tau,\tau')\left(\frac{N}{2}\widetilde{G}_x(\tau,\tau') - \sum_i^{N/2} \widetilde{\chi}_i^x(\tau)\widetilde{\chi}_i^x(\tau')\right)},
\tag{A7}
$$

and plug it into the action (A3). This allows us to integrate out the Majorana fields. Note that the projector $P(s, s')$ (see Eq. (15)) needs to be introduced during this procedure. This finally leads to Eq. (14) in the main text. Note that in the main text and following appendices we remove all the "~" to keep the notation concise.

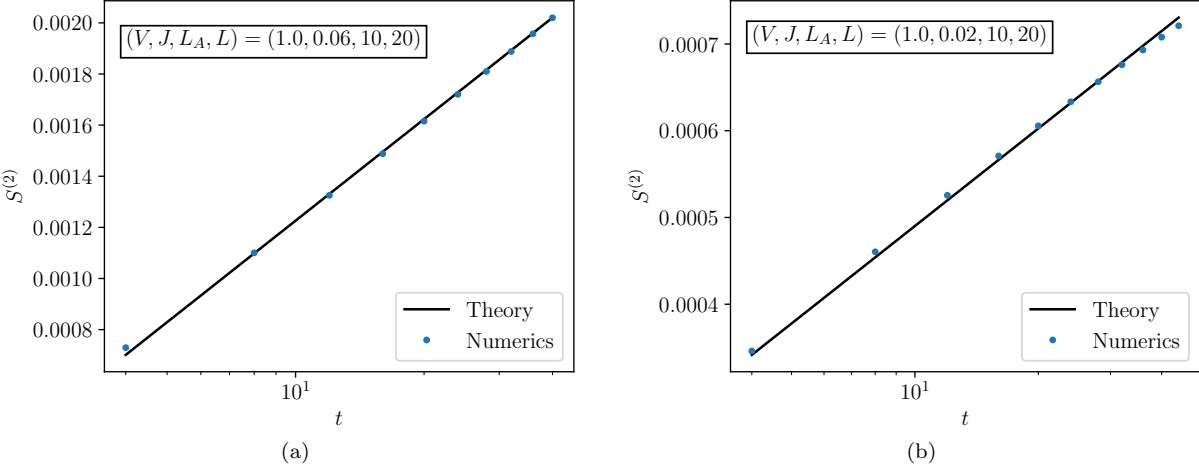

Figure 11. (a) $S^{(2)}$ as a function of $\log t$ in the $(4,4)$ model in the $J/V \ll 1$ limit. The theory result is drawn from Eq. (C2) with proper cutoff parameter $\epsilon$. (b) The same plot in the $(2,2)$ model in the $J/V \ll 1$ limit. The theory result is drawn from Eq. (C3) with proper cutoff parameter $\epsilon$.

## Appendix B: Replica symmetry, twist fields, and perturbative limit

One observation from Fig. (2) and the boundary conditions (A5) for $\chi_i^x$ is that if we redefine the fields by

$$\overline{\chi}_i^x(s) = \begin{cases} \chi_i^x(s+4t), & x \in A \text{ and } s \in [(4m+2)t, (4m+4)t), \\ \chi_i^x(s), & \text{otherwise} \end{cases}$$

then all the $\overline{\chi}_i^x$ carries the same boundary conditions that were original carried only by $\chi_i^x(s)$ with $x \in B$. Formally $\overline{\chi}_i^x(s)$ is related to $\chi_i^x(s')$ by some orthogonal matrix $\mathcal{A}(s,s')$ when $x \in A$[5]. This redefinition reveals some sort of replica "symmetry" in the partition function which allows to calculate the entropy in the small $J$ limit perturbatively. In the small $J$ limit, the saddle point solution for $\overline{G}_x^n$ is replica diagonal and is to leading order of $J$ well approximated by the $J = 0$ solution: $\overline{G}_x^n = 1_{n \times n} \otimes G_q$, where $G_q$ is the Green's function for a single-site $SYK_q$ cluster. These guarantee that $Z_n(t, L_A)$ looks almost exactly like $n$ decoupled copies of $Z(t)$, except for local terms at subsystem boundaries

$$\delta S = S_{n,\text{saddle}} - n S_{1,\text{saddle}}$$

$$= \frac{J^2}{4p} \sum_{x^*=0,L_A} \left\{ -\int_{[0,4nt]^2} ds ds' f(s) f(s') ((\mathcal{A}^T \overline{G}_{x^*}^n \mathcal{A})(s,s') \overline{G}_{x^*+1}^n(s,s'))^{\frac{p}{2}} P + n \int_{[0,4t]^2} ds ds' f(s) f(s') (G_{x^*}^1 G_{x^*+1}^1)^{\frac{p}{2}} P \right\},$$

$$(B2)$$

where $x^*$ are summing over $\partial A = \{0, L_A\}$, and we denoted the saddle point solution of $Z(t)$ by $G_x^1$.

Using the expressions of $\mathcal{A}$ and $P$, $\delta S$ can be simplified to

$$S^{(2)} = \frac{\delta S}{n-1} = \frac{2 \times 4n}{n-1} \frac{J^2}{4p} \int_0^t ds \int_t^{2t} ds' f(s) f(s') G_q^p(s-s'),$$

$$(B3)$$

where the prefactor $4n$ is the number of disjoint sectors (blocks) in the Green's functions on which the nonzero entries of the projected Green's functions $G_A P = \mathcal{A}^T \overline{G}_x^n \mathcal{A} P$ and $G_B P = \overline{G}_x^n P$ do not overlap.

---

[5] This orthogonal matrix is a version of the twist field; or in more rigorous sense, it implements the temporal branch cut that connect two twist fields at the boundary $x^*$. Viewing the $n$ Majorana fields from the $n$ replicas as separate fields, we can introduce the twist field $\mathcal{T}$ to incorporate the boundary condition between the $n$ fields, defined by

$$\mathcal{T}_{x^*}(t) \colon \chi_{i,m}^x(\tau) \to \chi_{i,m+1}^x(\tau), \quad x > x^*, \tau = t,$$

while $\mathcal{T}_{x^*}$ acts trivially on $\chi_{i,m}^x(\tau)$ otherwise. Then we have

$$\mathcal{T}_{x^*}^\dagger(t) \colon \chi_{i,m}^x(\tau) \to \chi_{i,m-1}^x(\tau), \quad x > x^*, \tau = t.$$

This way we have

$$Z_n(t) = \int_{\mathcal{C}} [D\chi_{i,m}^x] \mathcal{T}_{x^*+L_A}(4t) \mathcal{T}_{x^*+L_A}^\dagger(2t) \mathcal{T}_{x^*}^\dagger(4t) \mathcal{T}_{x^*}(2t) e^{-NS}$$

$$= \text{Tr} \left[ \mathcal{T}_{x^*+L_A} \mathcal{T}_{x^*}^\dagger \rho^{\otimes n} \mathcal{T}_{x^*+L_A}^\dagger \mathcal{T}_{x^*} \rho^{\otimes n} \right].$$

$$(B1)$$

## Appendix C: Perturbative result for the (4,4) and (2,2) models

In this Appendix we outline the main result for the $(p, q) = (4, 4)$ model and additional perturbative result for the $(2, 2)$ model. For the $(4, 4)$ model, the strong inter-cluster coupling limit $J \gg V$ shows similar behavior as the $(4, 2)$ model: both are described by Eq. (26). As one moves away from this limit, $S^{(2)}$ decays with increasing $V$ just as in the $(4, 2)$ model, however the decay is much slower and no drastic change of slope of $S^{(2)}(V/J)$ is observed in our numerics.

The weak coupling limit $J \ll V$ is more interesting, in which $S^{(2)}$ shows an early time behavior that scales linearly with $\log t$, see Fig. 11(a). This scaling can be understood from Eq. (B3). Using the conformal Green's function for $q = 4$

$$G_c(s, s') = \frac{1}{\left(2V\sqrt{\pi}\frac{2t}{\pi}\sin\frac{\pi(s-s')}{2t}\right)^{1/2}} \tag{C1}$$

and plug it into Eq. (B3), we have

$$S^{(2)} = \frac{J^2}{2\pi V^2}\log\frac{2t}{\pi\epsilon} - \frac{\pi J^2\epsilon^2}{24V^2t^2} + O\left((\epsilon/t)^3\right), \tag{C2}$$

The logarithmic $t$ behavior observed in Fig. 11(a) is then identified with the first term.

We now apply the same method to the $(2, 2)$ model in the weak coupling limit $J \ll V$, in which an early time $S^{(2)} \propto \log t$ behavior is also observed (see Fig. 11(b)). The scaling behavior can again be obtained by plugging the conformal Green's function (29) to Eq. (B3). Note that since for generic $q$ we have $G_c \propto \left(\sin\frac{\pi(s-s')}{2t}\right)^{-\frac{2}{q}}$ while the integrand in Eq. (B3) is $p$-power of $G_c$, we expect that all the $p = q$ model exhibits the same behavior (up to a numerical constant) for any value of $p = q$. For $q = p = 2$ we get

$$S^{(2)} = \frac{4J^2}{\pi^2 V^2}\log\frac{2t}{\pi\epsilon} - \frac{J^2\epsilon^2}{3V^2t^2} + O((\epsilon/t)^3), \tag{C3}$$

which indeed is the same as the (4,4) result Eq. (C2) up to a numerical prefactor.

To conclude, the early time behavior of $S^{(2)}$ at $J/V \ll 1$ for both models is in good agreement with the perturbative results (C2) and (C3). We emphasize that these perturbative results should only work for $V^{-1} \ll t \ll \Lambda$, where a large time cutoff $\Lambda$ needs to be introduced, since we know that $S^{(2)}$ must be bounded at long time. We also note that although early time $\log t$ behavior appears for both $J/V \ll 1$ and $J/V \sim 1$ regimes in the $(2, 2)$ model, they are still qualitatively different since the prefactor $\alpha \propto J^2/V^2$ in one regime while $\alpha \propto J/V$ in the other.

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
