# Peer review of "Non-unitary dynamics of Sachdev-Ye-Kitaev chains"

_SciPost Physics_

## Round 1 · Referee Report · Anonymous (Referee 1) · 2020-12-1

Report

In this work, the authors consider one-dimensional chains, where each local system is described by a SYK-like (non-Hermitian) Hamiltonian with p-body interactions and additional inter-site, q-body coupling terms. This type of chains (in the unitary case) has already received significant attention in the literature, since it represents a tractable chaotic model where different aspects of the many-body dynamics can be tackled analytically.

The authors study a protocol where the evolution is driven by a non-Hermitian Hamiltonian. The main motivation comes from recent studies where chaotic unitary evolution is combined with local measurements, leading to an entanglement-scaling phase transition.

The authors present detailed analytical and numerical calculations, supporting two main results: 1) the interacting model exhibits a first order phase transition from the volume-law entanglement phase to an area law phase; 2) in the non-interacting case (i.e. p=q=2) the authors find that the model displays an extensive critical regime with emergent two-dimensional conformal symmetry.

I believe the draft is very well written: the motivations are transparent, and the calculations and results are presented in a clear way, with useful paragraphs often summarizing the results. Furthermore, the research presented is certainly timely, and of interest for a broad audience. Finally, the results appear to be correct. For these reasons, I recommend publication, after the authors have addressed a few minor comments, that I list below.

a) First, I find that Sec. 3.2 could benefit from additional comments on the results presented. For example, I understand that Eq. (31) is, at present, a conjecture [differently from Eq. (32)], although very natural from the physical point of view and supported by numerics. Is this true? If not, the authors might comment on how this formula is derived. Analogously, I missed whether Eq. (33) can be proven, or it is suggested as a natural conjecture.

b) Regarding again the free case, do the authors expect that the critical phase remains for finite values of N? Or rather than a volume-law phase will emerge for some finite range of J/V? The authors might comment on this, since it is a natural and interesting question.

c) Since the phase transition discovered by the authors is first order, one would expect a signature of this in the coefficient s0 appearing, for instance, in Eq. (26). Namely, I would expect that as V/J approaches the phase transition from below, s0 approaches a finite value s different from 0 and smaller than \log(2)/2. Is it possible to estimate s from the analytical side?

d) Finally, a curiosity (which I don’t expect the authors to comment on in the draft). One could study a different evolution protocol, where the unitary dynamics is alternated with the non-unitary one during different time intervals (namely, V=V(t), J=J(t) are two alternating step functions). This protocol would mimic even more closely the measurement dynamics studied in different random-circuit protocols. Would similar techniques apply in this case, either in the Brownian version of the model, or for quenched disorder?

e) Typo in second-to-last line in page 8: “we provide an explain” “we explain”

  • validity: -
  • significance: -
  • originality: -
  • clarity: -
  • formatting: -
  • grammar: -

Author:  Chunxiao Liu  on 2021-01-13  [id 1142]

(in reply to Report 1 on 2020-12-01)
Category:
remark
answer to question
correction

We would like to thank the referee for his/her thoughtful comments and efforts towards improving our manuscript.

Referee's minor comments:

The referee writes:
“First, I find that Sec. 3.2 could benefit from additional comments on the results presented. For example, I understand that Eq. (31) is, at present, a conjecture [differently from Eq. (32)], although very natural from the physical point of view and supported by numerics. Is this true? If not, the authors might comment on how this formula is derived. Analogously, I missed whether Eq. (33) can be proven, or it is suggested as a natural conjecture.”

Our response:
Both Eqs. (31) and (33) are conjectures that are supported by numerics. We have rephrased the relevant sentences to more clearly reflect this point in the revised manuscript. (See “Numerical evidence suggests…” above Eq. (31) and “numerics (as shown in Fig. 10c) suggests that” above Eq. (33).)

The referee writes:
“Regarding again the free case, do the authors expect that the critical phase remains for finite values of N? Or rather than a volume-law phase will emerge for some finite range of J/V? The authors might comment on this, since it is a natural and interesting question.”

Our response:
We thank the referee for raising this interesting question. In Ref. [56], one of the authors (with other collaborators) explored the Brownian non-unitary free fermion dynamics at N=1. In this model, the randomness is uncorrelated in both spatial and temporal directions. They observed an emergent conformal symmetry in this model when the ratio J/V is finite. This result holds for any finite N and also the large N limit, which corresponds to the Brownian non-unitary free SYK model.
It is surprising/interesting to observe the same scaling behavior in the quenched non-unitary free SYK model in which the randomness is correlated in the temporal direction. For this model, we expect that at finite N, it may not have conformal symmetry anymore. We are currently working in this direction.

The referee writes:
“Since the phase transition discovered by the authors is first order, one would expect a signature of this in the coefficient s0 appearing, for instance, in Eq. (26). Namely, I would expect that as V/J approaches the phase transition from below, s0 approaches a finite value s* different from 0 and smaller than \log(2)/2. Is it possible to estimate s* from the analytical side?”

Our response:
We thank the referee for pointing out this question. We estimated the location of the first order phase transition by looking at the scaling of the type II saddle point (we also compared the two saddles at this point), however at this moment we are not aware of analytical approaches of extracting the value s* at this transition point.

The referee writes:
“Finally, a curiosity (which I don’t expect the authors to comment on in the draft). One could study a different evolution protocol, where the unitary dynamics is alternated with the non-unitary one during different time intervals (namely, V=V(t), J=J(t) are two alternating step functions). This protocol would mimic even more closely the measurement dynamics studied in different random-circuit protocols. Would similar techniques apply in this case, either in the Brownian version of the model, or for quenched disorder?”

Our response:
We thank the referee for bringing up this nice variant of our model. Our saddle point technique can be readily applied to this case (either the Brownian or the quenched version). We tried this model and we found that this variant agrees with that of our model.

The referee writes:
“Typo in second-to-last line in page 8: ‘we provide an explain’ ‘we explain’”

Our response:
We have corrected this typo in the revised draft.

---

## Round 1 · Referee Report · Anonymous (Referee 2) · 2020-12-8

Strengths

1 timely topics
2 overall solid method
3 interesting results

Weaknesses

1 analysis of numerical results a bit sketchy
2 presentation of methods could be clearer

Report

This manuscript studies how entanglement grows under a non-unitary dynamics. One motivation is the entanglement in open quantum systems, in particular, quantum trajectories with continuous measurement. Here the authors consider a dynamics generated by a non Hermitian Hamiltonian. While a roughly similar phenomenology is expected in both cases (competition between entangling unitary and disentangling non-unitary), the non-Hermitian case is not as physical yet somehow simpler.

Compared to a previous work on the same setup, the authors chose to study a large N system (an SYK chain), which allows to make the problem tractable, i.e., polynomially hard with numerics, even when the Hamiltonian is not free. The second Renyi entropy (which the authors focused on) is expressed as a difference of large N partition functions/functional integrals, and the saddle points are obtained by Schwinger-Dyson equations. The latter is not treated analytically in general, but solutions are looked for by iteration.

The authors considered two particular cases: an interacting one and a free one. The results on the free model is rather in agreement with a previous work. The interacting results have some interesting new features, including a volume-area law transition, and the existence of different types of saddles.

I have a few concerns on the way the authors get from the general methods to the results. The analysis a bit sketchy right now, and needs perhaps some sharpening. Here are some specific points to consider:

First, how does one determine which saddle dominates the partition functions in the Renyi entropy? In principle one can compute the action (over N) of the saddles and compare them. The non-analytical behavior at the transition will be also a more precise way to characterize the transition. I wonder why the authors did not look at that (and present it clearly).

The other concern is the numerical iteration method. In SYK-type models, it is often necessary to use a more “continuous” iteration scheme $G \to G_{old} (1-x) + x G_{new}$ (with x well below 1) to converge. Such scheme may also detect the (in)stability of saddles. Here, the authors only used an iteration scheme with $x = 1$ (however they chose several “initial conditions”). It would be nice to try different schemes to confirm the saddles, maybe study their stability and simplify the initial condition dependence.

Final question on numerics. Usually, real time dynamics in SYK models is simulated using the Kadanoff-Baym equation, which evolves forward in time in a causal way. Here the iteration scheme is quite different and is not causal. Is that necessary to get replica non-diagonal solutions? Can the authors comment on that?

Some issues on presentation: - The separation of even and odd Majorana modes is a technical “trick” to implement the initial condition. It might be obvious to the authors, but I found it not clearly explained (neither in the main text nor in the appendix), and some readers could be lost right at eq (9). - Related: usually, the $n$-th Renyi entropy gives rise to a $2n$-fold contour. Here, we have a $4n$-fold one, which is due to the even-odd separation. It’s better to point this out in the main text for conceptual clarity. - The “initial conditions” in the numerical details section are initial conditions for the Schwinger-Dyson iteration, and do not correspond to $t = 0$. So the term “initial condition” can be confusing. - What is the status of the point $V = 0.3$? Is it just a crossover? The "phase diagram" looks like there are two transitions, which is not the authors wanted to claim.

Overall this is an interesting work (although a bit formal), and I would recommend its publication after the above concerns are addressed.

  • validity: -
  • significance: -
  • originality: -
  • clarity: -
  • formatting: -
  • grammar: -

Author:  Chunxiao Liu  on 2021-01-13  [id 1143]

(in reply to Report 2 on 2020-12-08)

We would like to thank the referee for his/her thoughtful comments and efforts towards improving our manuscript.

The referee writes:
“First, how does one determine which saddle dominates the partition functions in the Renyi entropy? In principle one can compute the action (over N) of the saddles and compare them. The non-analytical behavior at the transition will be also a more precise way to characterize the transition. I wonder why the authors did not look at that (and present it clearly).”

Our response:
When there are multiple saddle point solutions, the Renyi entropy is dominated by the saddle that gives the smallest action (the physical saddle). At every time step we followed this procedure and calculated the action corresponding to different saddle points and determined the physical saddle. By our definition, the transition happens when the steady state switches from a volume law state to an area law state (See Fig. 8(b) for comparison of different saddle point solutions and the non-analytical behavior around the transition point).

The referee writes:
“The other concern is the numerical iteration method. In SYK-type models, it is often necessary to use a more “continuous” iteration scheme G→Gold(1−x)+xGnew (with x well below 1) to converge. Such scheme may also detect the (in)stability of saddles. Here, the authors only used an iteration scheme with x=1 (however they chose several “initial conditions”). It would be nice to try different schemes to confirm the saddles, maybe study their stability and simplify the initial condition dependence.”

Our response:
We thank the referee for pointing out this question. In fact we precisely followed this iteration method with an ρ as in “G→Gold(1− ρ)+ ρ Gnew” often chosen to be 0.5. We also varied the initial seed Green’s functions in order to do an extensive search for and test the stability of various saddle points. We apologize for not presenting this clearly in the manuscript and have now added a sentence “…according to $G^{(n)}_x=(1-\rho) G^{(n-1)}_x+\rho G^{(n-1)}_{x,\text{new}}$, where $0<\rho<1$ is some weight that controls the rate of convergence …” above Eqs. (22) to clarify this issue in the revised manuscript.

The referee writes:
“Final question on numerics. Usually, real time dynamics in SYK models is simulated using the Kadanoff-Baym equation, which evolves forward in time in a causal way. Here the iteration scheme is quite different and is not causal. Is that necessary to get replica non-diagonal solutions? Can the authors comment on that?”

Our response:
While the Kadanoff-Baym equations for real time Green’s function is causal, the saddle point equations to compute Renyi entropy is not causal: because of the two-step trace structure $\mathrm{TR}_A(\mathrm{Tr}_B \rho)^n$, the forward time evolution and backward time evolution do not cancel and the saddle point equations must be solved consistently along the entire contour. This is crucial in obtaining a replica non-diagonal solution which is called a replica wormhole solution.

The referee writes:
“Some issues on presentation:
The separation of even and odd Majorana modes is a technical “trick” to implement the initial condition. It might be obvious to the authors, but I found it not clearly explained (neither in the main text nor in the appendix), and some readers could be lost right at eq (9).”

Our response:
We have added a few sentences (above and below Eq. (9)) in the revised manuscript to clarify this point.

The referee writes:
“Related: usually, the n-th Renyi entropy gives rise to a 2n-fold contour. Here, we have a 4n-fold one, which is due to the even-odd separation. It’s better to point this out in the main text for conceptual clarity.”

Our response:
We have added the sentence “…note also that the contour sews even and odd Majorana fields …” below Eq. (11) in the revised manuscript to clarify this point.

The referee writes:
“The “initial conditions” in the numerical details section are initial conditions for the Schwinger-Dyson iteration, and do not correspond to t=0. So the term “initial condition” can be confusing.”

Our response:
We have added the sentence “for numerical iteration (not to be confused with the Green's function solution at $t=0$)” above Eq. (24) in the revised manuscript to clarify this point.

The referee writes:
“What is the status of the point V=0.3? Is it just a crossover? The "phase diagram" looks like there are two transitions, which is not the authors wanted to claim.”

Our response:
When V<0.39, the entanglement entropy of the steady state has volume law scaling. The V=0.3 point marks two different early time behaviors. The early-time non-monotonic behavior (first increase and then decrease) in the regime 0.3<V<0.39 has also been observed in many non-unitary dynamics with a small N limit. We have added a few sentences at the end of Sec. 3.1.3 (see “In the regime 0.3 < V < 0.39…”) in the revised draft to clarify this point.

The referee writes:
“Overall this is an interesting work (although a bit formal), and I would recommend its publication after the above concerns are addressed.”

Our response:
We would like to thank the referee for carefully reading our manuscript and recommending our work for publication in SciPost.

---

## Editorial Decision

resubmitted